# No Difference in the Prevalence of HIV-1 *gag* Cytotoxic T-Lymphocyte-Associated Escape Mutations in Viral Sequences from Early and Late Parts of the HIV-1 Subtype C Pandemic in Botswana

**DOI:** 10.3390/vaccines11051000

**Published:** 2023-05-19

**Authors:** Baitshepi Mokaleng, Wonderful Tatenda Choga, Ontlametse Thato Bareng, Dorcas Maruapula, Doreen Ditshwanelo, Nametso Kelentse, Patrick Mokgethi, Natasha Onalenna Moraka, Modisa Sekhamo Motswaledi, Leabaneng Tawe, Catherine Kegakilwe Koofhethile, Sikhulile Moyo, Matshediso Zachariah, Simani Gaseitsiwe

**Affiliations:** 1Botswana Harvard AIDS Institute Partnership for HIV Research and Education, Gaborone 999106, Botswana; bmokaleng@bhp.org.bw (B.M.); wtchoga@gmail.com (W.T.C.); barengntlame@gmail.com (O.T.B.); maruapulad@bhp.org.bw (D.M.); ditshwanelod@bhp.org.bw (D.D.); thobanenametso@gmail.com (N.K.); pmokgethi@bhp.org.bw (P.M.); natasha.o.moraka@gmail.com (N.O.M.); khei79@gmail.com (C.K.K.); smoyo@bhp.org.bw (S.M.); 2School of Allied Health Professions, Faculty of Health Sciences, University of Botswana, Gaborone 999106, Botswana; motswaledims@ub.ac.bw (M.S.M.); 200804209@ub.ac.bw (L.T.); zachariahm@ub.ac.bw (M.Z.); 3Department of Biological Sciences, Faculty of Science, University of Botswana, Gaborone 999106, Botswana; 4Department of Immunology and Infectious Diseases, Harvard School of Public Health, Boston, MA 02115, USA

**Keywords:** HLA-associated, *gag*, recent infections, CTL escape mutations, HIV evolution, Botswana

## Abstract

HIV is known to accumulate escape mutations in the *gag* gene in response to the immune response from cytotoxic T lymphocytes (CTLs). These mutations can occur within an individual as well as at a population level. The population of Botswana exhibits a high prevalence of HLA*B57 and HLA*B58, which are associated with effective immune control of HIV. In this retrospective cross-sectional investigation, HIV-1 *gag* gene sequences were analyzed from recently infected participants across two time periods which were 10 years apart: the early time point (ETP) and late time point (LTP). The prevalence of CTL escape mutations was relatively similar between the two time points—ETP (10.6%) and LTP (9.7%). The P17 protein had the most mutations (9.4%) out of the 36 mutations that were identified. Three mutations (A83T, K18R, Y79H) in P17 and T190A in P24 were unique to the ETP sequences at a prevalence of 2.4%, 4.9%, 7.3%, and 5%, respectively. Mutations unique to the LTP sequences were all in the P24 protein, including T190V (3%), E177D (6%), R264K (3%), G248D (1%), and M228L (11%). Mutation K331R was statistically higher in the ETP (10%) compared to the LTP (1%) sequences (*p* < 0.01), while H219Q was higher in the LTP (21%) compared to the ETP (5%) (*p* < 0.01). Phylogenetically, the *gag* sequences clustered dependently on the time points. We observed a slower adaptation of HIV-1C to CTL immune pressure at a population level in Botswana. These insights into the genetic diversity and sequence clustering of HIV-1C can aid in the design of future vaccine strategies.

## 1. Introduction

Botswana has the third highest HIV rate in the world, with a prevalence of 18.6% among the population aged 15–49 years, despite the provision of free combination antiretroviral therapy (cART) to people living with HIV (PLWH) [1]. Strict adherence to cART lowers HIV viremia, ceases progression to acquired immune deficiency syndrome (AIDS), and reduces the risk of transmission [2]. However, its efficacy is threatened by poor medication access, poor medication adherence, and emergence of drug resistance mutations (DRMs) [3,4]. Due to the aforementioned factors, cART alone cannot end the HIV-1 epidemic; thus, an effective vaccine may contribute to reducing or ending the epidemic.

Cytotoxic T lymphocytes (CTLs) have been shown to play a major role in controlling viremia and the disease progression of HIV-1 infections [2,5,6]. HIV replication can be controlled by immune pressure from CTLs as shown among treatment-naïve elite controllers [7]. Studies have reported that CD8+ T cells reduce the viral load (VL) set points which hold back viral replication, resulting in the slowing down of the disease progression [2,8]. However, HIV-1 is one of the most diverse and rapidly evolving human pathogens due to its error-prone reverse transcriptase mechanism, leading to the development of variants that cannot be recognized by initial CTL responses, a process termed CTL escape [9]. These variants occur around the processing and presentation of viral epitopes to the human leucocyte antigen (HLA). CTL immune responses are abrogated by CTL escape mutations, and variants with these mutations have a competitive advantage and can be transmitted and, therefore, spread at the population level [2,10,11].

There is evidence from different regions, including sub-Saharan Africa, that HIV can adapt to HLA class I alleles at the population level [9,10]. HLA-A30, A02, A23, A68, B58, B72, B42, B8, B18, B44, B45, Cw7, Cw2, Cw17, Cw6, and Cw4 have been associated with different disease progressions in several populations, and HLA-A30, A02, A23, A68, B58, Cw2, Cw4, Cw6, Cw7, and Cw17 are the most frequent (>10%) types in Botswana [12,13]. HIV-1 evolves rapidly within individuals by developing CTL escape mutations [9]. However, CTL escape mutations can persist upon transmission to an HLA-mismatched host and become the dominant variants in a population, especially if the mutations do not carry a fitness cost or if they are accompanied by compensatory mutations [14]. In addition, understanding how the virus evolves within a population may improve treatment modalities and aid towards the development of effective drugs and vaccine development.

We hypothesized that HIV-1C is adapting to the Botswana population, especially to the dominant HLA class I alleles in the population, which influences HIV-specific CTL responses. This would be evidenced by HIV-1 variants in recently infected individuals evolving to accumulate CTL escape mutations over time. If this is the case, this would be important to ascertain, as this would inform HIV vaccine design, especially for CTL-based vaccines. So, here we sought to determine the prevalence of HIV-1C CTL escape mutations in viruses from the earlier part of the epidemic and compare that with the prevalence of these mutations in the later part of the HIV-1 epidemic in Botswana. Besides informing vaccine design, data from this project will also add to the knowledge of viral evolution in relation to immune responses.

## 2. Materials and Methods

This study utilized a total of 196 PLWH who are ART-naïve from 3 independent cohorts, namely Tshedimoso, Tlhotlhomiso, and the Botswana Combination Prevention Project (BCPP) from Botswana, which are previously described elsewhere [15,16]. Tlhotlhomiso cohort was designed to evaluate the presence of transmitted drug resistance (TDR) in treatment naïve PWH in Botswana, described elsewhere [17], we screened samples for recency of infection which was included in this study. Participants were recruited at antenatal clinics where women between the ages of 18 and 25 years in their first pregnancy were enrolled.

### 2.1. Selection of the Study Samples and Determination of HIV Recent Infection

Samples from individuals who were identified as recently infected were utilized in the study. A total of 196 individuals who were recently infected were further classified into 2 time points, which were the early time point (ETP) (2004 to 2008) and the later time point (LTP) (2015 to 2018). A total of 42 sequences from the Tshedimoso study were used as the ETP, and 154 sequences from Tlhotlhomiso (Appendix A) and the BCPP study were used as the LTP. Briefly, the seroconversion for the 8 sequences of the 42 acutely HIV-1 infected individuals was estimated as the midpoint between the last seronegative test and the first seropositive seroconversion. For the remaining 34 sequences, the recency was estimated by the Fiebig stage assignment described elsewhere [18]. A detailed genotyping method has been previously described [16,19]. All of the 42 *gag* sequences were retrieved from the Los Alamos HIV database under accession numbers GQ27538–GQ277569, GQ375107–GQ375128, and GQ87074–GQ871183 (Appendix A).

Of the 154 sequences, 140 were generated from PLWH who were enrolled in the BCPP [20]. Originally, the BCPP study included 12,610 participants recruited from 30 communities, and 140 participants were identified as recent infections. Two independent approaches were used to identify recent infections. Firstly, during enrolment, participants who tested HIV-positive with the HIV rapid test were screened for recent infection using the LAg avidity EIA SEDIA serological assay (Sedia Biosciences Corporation, Portland, OR, USA) (n = 31). Secondly, 109 ART-naïve participants who tested HIV-1-negative at the first visit and HIV-positive on the second visit after a year were considered as recently infected (Appendix A).

### 2.2. Ethical Statement and Consent Process

All studies were approved by the Health Research and Development Committee (HRDC) of the Republic of Botswana (HPDME 13/18/1) and the Office of Human Research Administration (OHRA) of the Harvard T.H. Chan School of Public Health for the Tshedimoso and Tlhotlhomiso study and the US Centers for Disease Control and Prevention (Protocol 6475) for the BCPP. These studies were conducted in accordance with the principles in the declaration of Helsinki. Participants included in this study signed a consent form for their samples to be used in future studies.

### 2.3. HIV-gag Genotyping (Extraction, Amplification and Sequencing)

#### 2.3.1. HIV-1 RNA Extraction and Amplification Used to Generate N_1_ Sequences

A total of 14 sequences from Tlhotlhomiso were sequenced from previously stored plasma samples. Firstly, HIV RNA was extracted using the EZ1 Advanced XL Machine (Qiagen, Hilden, Germany). The HIV-1 *gag* was amplified with reverse transcriptase PCR (RT-PCR) using the Transcription First-Strand cDNA Synthesis kit protocol (Roche One-Step Diagnostics, Indianapolis, IN, USA). The 25 µL reaction master mix for each sample was composed of 10 µL of the viral RNA, 5 µL of the 5× buffer, 0.5 µL of the enzyme Roche One-Step, 1.25 µL of 10 nm for each primer (forward and reverse), and 7 µL of water. The RT-PCR primer sequences used include F2NST (5′-GCGGAGGCTAGA AGGAGAGAGATGG-3; HXB2 nt: 769–793) and 1448L (5′-AGGGGTCGCTGCCAAAGAGTGATT-3′; HXB2 nt: 2258–2291). The RT-PCR cycling conditions were performed as follows: reverse transcription at 50 °C for 30 min, initial denaturation at 94 °C for 7 min, followed by 10 cycles of the second denaturation at 94 °C for 10 s, annealing at 55 °C for 30 s, and extension at 68 °C for 2 min. This was followed by 35 cycles of denaturation at 94 °C for 10 s, an annealing temperature of 53 °C for 30 s, and a final extension at 68 °C for 2 min with a hold stage at 4 °C for a maximum of 18 h. PCR products were viewed in 1.0% gel electrophoresis using a 1 kb DNA ladder. Amplicons whose band size (~1.5 kb) corresponded to the right size were proceeded for sequencing. Faint first-round PCR bands and those that failed amplification were subjected to a second round of PCR, whereby primers *gag*-5U (5′-GTGCGAGAGCGTCAATATTAAGAG-3′; HXB2 nt: 794–817 and 1445L) and (5′-GGTCGCTGCCAAAGAGTGATT-3′; HXB2 nt: 2258–2278) were used with the Phusion DNA polymerase kit (New England Biolabs, France). The reaction mixture for each sample was 25 µL, comprising 9.0 µL of sterile water, 12.5 µL of 2× Buffer, 1.25 µL of primer *gag*-5U, 1.25 µL of primer 1445 L, and 1 µL of the RT-PCR product from the first-round PCR product. Thermal cycling conditions involved initialization at 95.5 °C; 35 cycles consisting of denaturation at 98 °C for 10 s, annealing at 62 °C for 30 s, extension at 72 °C for 20 s, and a final elongation at 72 °C for 10 min; and, finally, the products were kept at a hold stage at 4 °C.

#### 2.3.2. Sequencing of *gag*

Purification of the amplicons was performed using the QIAquick PCR Purification kit (Qiagen Inc., Valencia, CA, USA) according to manufacturer’s instructions. Amplicons were used to carry out cycle sequencing using BigDye chemistry (Big Dye terminator sequencing kit) according to manufacturer’s instructions (Applied Biosystems, Foster City, CA, USA). The ZR DNA sequencing clean-up kit (Zymo Research, Pretoria, South Africa) was used to purify the cycle sequencing product, and then the samples were loaded in a 96-optical reaction plate on the ABI prism 3130XL genetic analyzer (Applied Biosystems, Foster City, CA, USA) for generation of HIV *gag* sequences.

### 2.4. Hypermutation Screening in BCPP Participants

Hypermutations were screened from proviral DNA sequences generated using the hypermut tool imbedded in the Los Alamos webtools (https://www.hiv.lanl.gov/content/sequence/HYPERMUT/hypermut.html/, accessed on 28 August 2020). The HIV-1 C consensus sequence was used as a reference. The adjusted hypermutations were accounted for using the cumulative number of mutations across the length of the sequences from the analyzed HIV-1 *gag* gene. Sequences with a *p*-value of 0.05 were considered as hypermut. The adjustment for hypermutations was performed before the CTL escape mutations analysis for quality control. Hypermutations were adjusted at nucleotide position level, whereby only CTL mutations that were not associated with hypermutations were counted for the prevalence of CTL escape mutations.

### 2.5. Chromatograms and Data Sorting

Genetic analyzer output chromatograms were visualized and manually edited using Sequencher version 5.0 (Gene codes corporation, Ann Arbor, MI, USA). The sequences were trimmed at the beginning and the end to remove the ambiguous nucleotides, and sequences of high quality (quality score ≥ 69%) were assembled into contiguous sequences (contig). Mixed bases or ambiguous nucleotides were corrected based on the clean sequences that covered the same position. The final consensus sequence for each sample was exported in FASTA file format and further processed in Aliview ver. 1.26. Multiple sequence alignment (MSA) using HXB2 reference was performed using FastTree, and the final alignment was translated to corresponding amino acids for downstream analyses. The translated alignment was then partitioned based on time points.

### 2.6. Determining Cytotoxic T Lymphocyte Escape Mutations and Statistical Analysis

The 196 HIV *gag* sequences were analyzed for the known mutations associated with CTL escape documented in Table 1 [7,21,22,23,24,25,26,27,28,29,30,31,32,33,34,35,36,37,38,39,40,41,42,43,44,45,46,47,48,49,50,51,52]. The CTL escape mutations were analyzed per HIV *gag* regions in two time points according to Figure 1. Mutations were reported as percentages. Differences in prevalence of CTL escape mutations between the 2 time points were compared using a comparison of proportion test. All of the analysis was performed using STATA version 15. *p*-values less than 0.05 were considered statistically significant.

### 2.7. Phylogenetic Tree

The edge-trimmed MSA of HIV *gag* sequences (length of 786 bp nt long) was used to infer tree topology-based performing maximum likelihood (ML) analyses, and the bootstrap values were set at 1000. ModelTest v.3.7 [53] was used to select the simplest evolutionary model that adequately fit the sequence data. ML trees were implemented using IQTREE [54]. The Seaview tool was used to assess the architectures of the produced trees, and the final tree with timed outliers was finally visualized and annotated in Figtree v1.4.3. Posterior probabilities above 0.80 were noted as statistically significant. The pairwise distance among the sequences was assessed to determine their uniqueness as part of the quality control.

**Table 1 vaccines-11-01000-t001:** Documented CTL escape mutations.

Protein	Escape Mutation	Significance of Mutations	Reference
P17	K18R	Elicit lytic response	[21]
	R20K		[22]
	K26R	Reverting mutation	[23]
	H28Q		[24]
	K28R		[24]
	M30R		[25]
	Y79F		[26]
	Y79H	Diminished HLA binding	[27]
	A83T		[28]
	E93K	No CTL response detected	[29]
P24	V128G	Diminish CTL response	[30]
	A146P	Affect peptide processing	[31]
	I147L	Impair replication capacity	[7]
	I147M	Escape from epitope	[32]
	A163G	Affect presentation by HLA-I	[33]
	A165N	Compensatory mutation to A163G	[34]
	E173E	Compensatory mutation	[55]
	Q177D		[26]
	Q182T	Not recognized by CTL at all	[36]
	Q182E	Reduces viral replication capacity	[37]
	Q182G		[37,38]
	Q182S		[32,39]
	T186S		[32,39]
	V215L	Loss of viral fitness	[40]
	H219Q	Compensate deleterious effect caused by T242N	[41]
	I223V		[42]
	M228L		[42]
	T242N	Affect presentation by HLA-I	
	I247V		[43]
	G248A	In combination with T242N affect presentation by HLA-l	
	G248Q		[44]
	G248T	Reduces viral load on HLA-57/58 positive	[45]
	G248D		[45]
	A248G		[46]
	D260E		[47]
	R264K		[44]
	L268M	Affect recognition by TCR	[48]
	K302R		[39]
	T310S		[49]
	D312E	An escape to HLA-B 5801	[50]
	T332N		[51]
	G357S		[52]

HLA—human leucocyte antigen; CTL—cytotoxic T lymphocytes; TCR—T cell receptor.

## 3. Results

### 3.1. Characteristics of ETP and LTP HIV-1 gag Isolates

A total of 196 HIV-1 C *gag* sequences from PLWH in Botswana were included in the analysis. The data were made up of two sets; the ETP (n = 42) and LTP (n = 154) samples were collected in 2004–2008 and 2012–2018, respectively. The demographics are summarized in Table 2.

### 3.2. Comparing the Diversity of Historical and Recent HIV Isolates

Based on the Akaike Information Criterion (AIC), the best model was FLU + G4, and the Bayesian Information Criterion (BIC) was FLU + R5. A total of 146 HIV-1 C *gag* sequences from PLWH in Botswana were included in the analysis based on IQTREE filtering. The data were made up of two sets of ETP and LTP samples; ETP (n = 28) were collected in 2004–2008, and LTP (n = 117) in 2012–2018. The ML tree performed with long (786 bp) sequence lengths shows independent clustering of the two timepoints (Figure 2). When we performed the ML tree using proteins with shorter sequence lengths, such as P17 (HIV *gag* (nt position 130–1413); thus, ETP (n = 41) and LTP (n = 28)), we observed genetic intermixing of sequences from two time periods. The mean pairwise distances were 0.096 and 0.106 (*p* = 0.89) for ETP and LTP, respectively (Appendix A).

### 3.3. CTL Escape Mutations in gag Proteins

The first partition was nearly a full P17 gene (aa positions 15–133), with 41 ETP sequences and 28 LTP sequences. A total of 11 CTL escape mutations were observed within the P17 protein, as shown in Figure 3. Out of the 11 CTL escape mutations, 3 CTL escape mutations (K18R, Y79H, and A83T) were only found in the ETP. Although there was no statistical difference observed, mutations V128G, H28R, H28Q, E93K, and Y79F were predominant in the ETP, while M30R was predominant in the LTP. The partial HIV-1 C *gag* P24 protein (aa 133–192) was partitioned into 2 parts. In the first partition (P24), covering *gag* aa positions 134–192, 41 sequences were from the ETP, while and LTP had 34 sequences. In the *gag* aa positions 134–192, 11 CTL escape mutations were found. Mutations T190V (3%) and E177D (6%) were only reported in the LTP, while T190A (5%) was observed in the ETP only. The list of mutations reported in both time points was: I147M, Q182T, Q182S, T186S, A146P, S165N, A163G, and I147L, and no statistical difference was found among the time points (Figure 4a).

The second partition, (P24), covered *gag* aa positions 193–363. Totals of 41 and 151 sequences were analyzed for ETP and LTP, respectively. Thirteen CTL escape mutations were observed in P24. Mutations G248D, R264K, and M228L were observed only in sequences of the LTP. A total of nine mutations were reported in both time points. Out of these nine mutations, K331R was statistically higher in the ETP (10%) compared to the LTP (1%) *p* < 0.01), while H219Q was higher in the LTP (21%) compared to the ETP (5%) (*p* < 0.01). K302R remained constant over the two times points (Figure 4b).

Lastly, we used PROVEAN to observe CTL escape mutations that have not been reported, and predictions of whether they affect the HIV-1 *gag* have not been made. There were 27 observed variants that were significantly higher in the ETP, and 12 out of 27 were predicted as deleterious, as shown in Appendix A.

## 4. Discussion

The present study provides an insight into HIV-1 C *gag* gene evolution in the population of Botswana, which has documented high frequencies of HLA*B57/B58 alleles, between the two time points 10 years apart with ETP (2004–2008) and LTP (2014–2018) sequences. We report similar prevalence of CTL escape mutations in the HIV-1C *gag* sequences from the two periods. Our findings indicate that there is a slow accumulation of CTL escape mutations at the population level, suggesting that HIV is evolving at a slower rate in Botswana’s population. Our study utilized sequences from recently (within one year of infection) HIV-infected individuals to identify CTL escape mutations that are harbored by the transmitted virus and then selected by the host HLA. CTL escape mutations are cumulative within a population and persist upon transmission to an HLA-unmatched host and become the dominant variants in the population [31]. The CTL escape variants have been found to be less fit compared to the wild type, but they have the advantage of escaping the immune response and will gradually increase as the epidemic progresses [31]. When the less fit viruses are transmitted to new human host with a discordant HLA, they may have a clinical benefit, but if the host has the same HLA, then the recipient will fail to mount effective CTL immune responses [56,57,58]. On the other hand, the virus with deleterious mutations (e.g.,T242N in the TW10 epitope) becomes less fit and can develop compensatory mutations to either wholly offset the fitness cost the virus incurs (e.g., H219Q, I223V, and M228L) or partially restore the viral replication capacity (VRC) (e.g., I147L and S165N compensated for A146P in the ISW9 epitope and A163G in the KF11 epitope, restricted to B *5703/5801, respectively) [31,59,60]. In our findings, the T242N, TW10 epitope (TSTQEQIAW, *gag* aa:240–249) was found to be increasing with time, although it was not statistically significant; however, its compensatory H219Q was predominant in the LTP. Subsequently, the fully restored virus attains that of a wild type and will lead to disease progression.

A comparative study that aimed to investigate how HIV has evolved over time in Botswana and South Africa showed a strong positive correlation between HIV-1 C strains and HLA-B*57 and HLA-B*5801 alleles in Botswana, as opposed to South Africa [35]. This adaptation was associated with reduced VRC, rendering the virus less virulent [61]. A previous study by Payne et al. used sequences in the ETP and showed that HLA-B*57:02/57:03/58:01 were losing protectiveness in Botswana [25,35]. Theoretically, viral dynamics suggest that replicative fitness attenuates over time because of introduction of deleterious mutations within epitopes in order to evade the associated immune pressure. The accumulation of these escape mutations leads to a loss of epitopes (e.g., TW10) over time, a process termed epitope decay. In a study conducted in Japan (1983), individuals lacking the HLA-B*51 allele exhibited a 21% prevalence of the I135X mutation. Between 1997 and 2008, the frequency of the I135X mutation increased substantially, with accumulation observed in up to 70% of HLA-B*51-negative individuals [31]. Unexpectedly, our study shows that the ETP has a similar overall prevalence of CTL escape mutations to the LTP, suggesting a slow accumulation of CTL escape mutations over the two. Our study further investigated clustering of the sequences between time periods. Overall, we observed defined clustering of the *gag* sequences between the two time periods, suggesting different times of introduction of viruses into the population (time to most recent common ancestor: TMRCA). Nevertheless, we observed little genetic evolution between the two periods. This was also supported by similar mean DNA distance matrices (*p* = 0.89). When we investigated the clustering based on different proteins, such as P17, the clustering disappeared as expected. Thus, high coverage sequences are more desired when constructing phylogenetic inferences [62,63]. Our findings are also similar to those of Arien et al., a study that was conducted using fragments of HIV *pol* and *env* [64].

Based on the already documented CTL escape mutations, there is slow HIV evolution in the *gag* region [58,65]. However, we also observed escape variants that are statistically different among the two time points, but they were not classified in the Los Alamos HIV database. However, the PROVEAN prediction analysis shows that these escape CTL mutations have an impact on HIV *gag* function. This shows that HIV is evolving and adapting to the host immune system. The capacity of HIV-1 to escape from CTL responses threatens the effectiveness of CTL-mediated control of virus replication and poses a significant challenge for the development of effective vaccine-induced CTL responses [62,63,64]. Our findings warrant the further investigations based on in-silico analyses of CTL escape mutations to identify those that are associated with a high fitness cost. Although low viral escape has been documented due to low pressure [65,66], we postulate that our findings may have been influenced by sampling strategies and different methods of generating sequences or suggest evidence of polygenic adaptation to HIV-1C infection in Botswana (gains in HLA protectiveness). The present trends further attest to the need to molecularly characterize HLA alleles among PLWH currently in Botswana, particularly the elite controllers. Mechanisms responsible for an HIV-1 attenuation involve directional evolution towards increased survival and transmission at fitness cost; these findings can probably suggest that HIV-1 strains predominantly circulating in Botswana have evolved and excessively lost VRC to avoid CTL and other immune responses [58,64,67]. Despite our study only focusing on the HIV *gag* protein, our study provides important data on the HIV-transmitted variants that were circulating in two time points that are 10 years apart; however, this warrants further investigations on other HIV proteins.

In conclusion, we have investigated the HIV-1 evolution over time in Botswana, and no statistical difference was observed in CTL escape mutations among both the ETP and the LTP 10 years apart. These findings show that it is evident that HIV evolves at a slower rate in a population level. However, these findings open new avenues for additional research to investigate HIV-1 evolution in Botswana, such as the role of population genetics.

## Figures and Tables

**Figure 1 vaccines-11-01000-f001:**
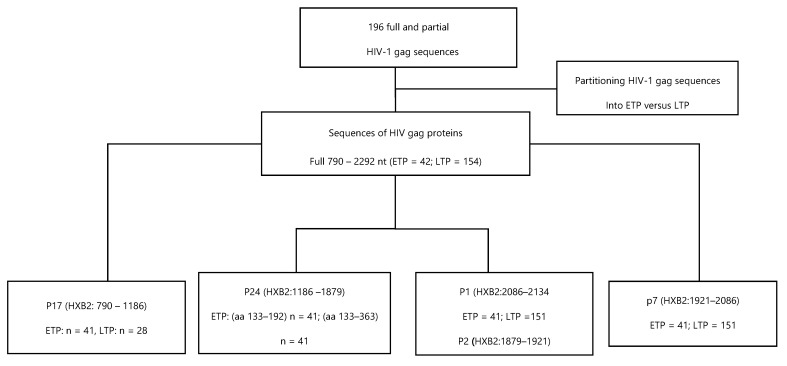
Schema showing how sequences were partitioned to perform the post-mutation analyses.

**Figure 2 vaccines-11-01000-f002:**
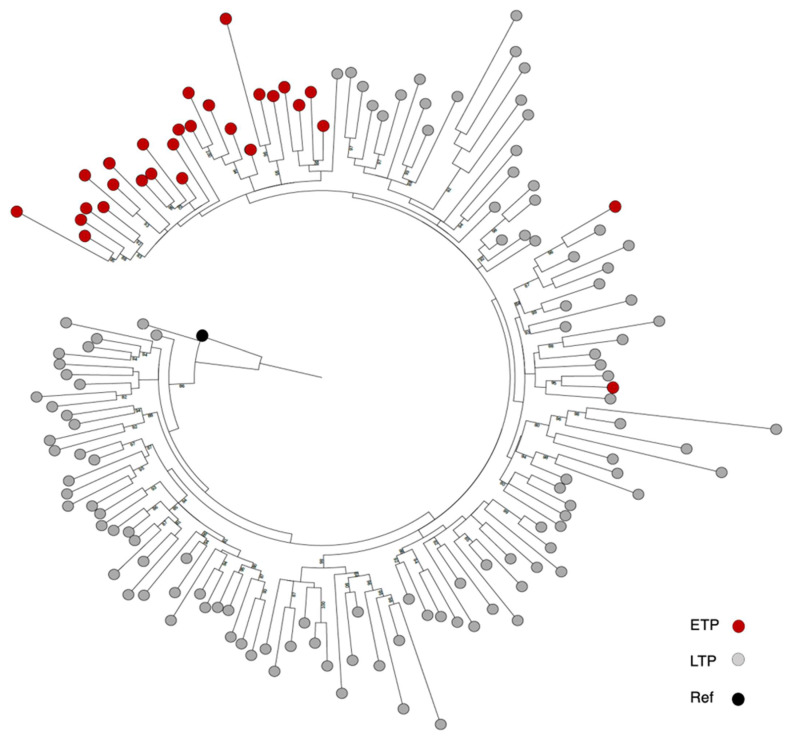
Phylogenetic tree of the near full-length HIV-1C *gag* sequences from the two time periods of the HIV-1 epidemic in Botswana. Maximum likelihood method based on near complete (nt = 786 base-pairs).

**Figure 3 vaccines-11-01000-f003:**
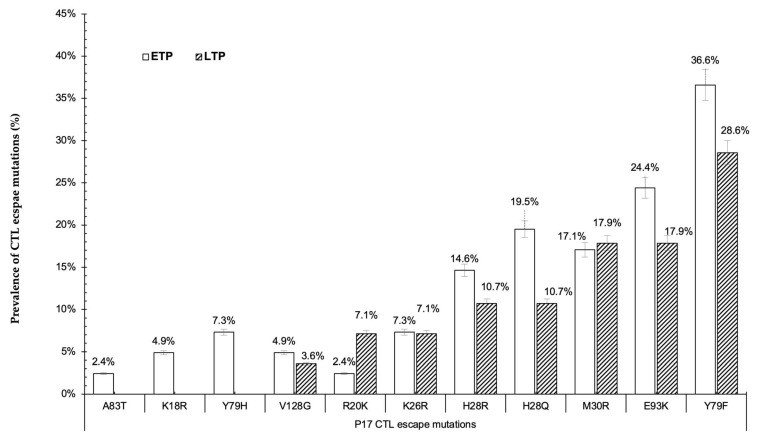
Prevalence of CTL escape mutations among sequences from ETP and LTP of HIV pandemic in Botswana.

**Figure 4 vaccines-11-01000-f004:**
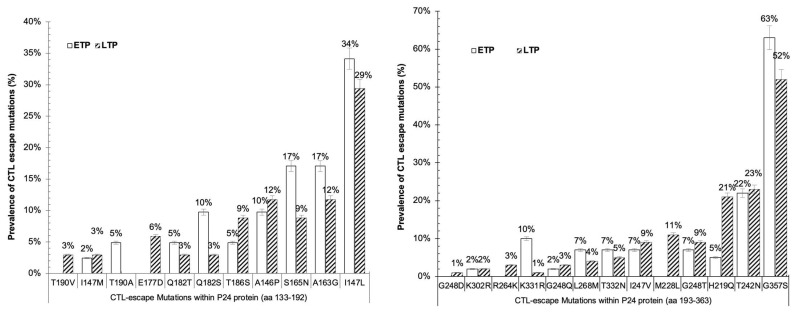
Prevalence of CTL escape mutations among sequences of P24 from ETP and LTP of HIV pandemic in Botswana.

**Table 2 vaccines-11-01000-t002:** Characteristics of the participants’ samples by ETP and LTP.

		ETP (N = 42)	LTP (N = 154)
Time of sample collection		2004–2008	2012–2018
Sample used to isolate HIV-1 isolates		Plasma	Buffy coat + plasma
HIV-1 subtype		C (42)	C (154)
Distance matrices		0.096	0.106
Mean viral load (VL) (copies/mL) (*p*-value < 0.015)		31,400 (IQR: 3356–209,500)	11,152 (IQR: 2475–33,349)
Mean time to seroconversion (days)		Duration of recent infection of approximately 130 days	Duration of recent infection of approximately 130 days
Gender distribution		9 Males, 33 Females	28 Males, 126 Females
Age (median; Q1, Q3) years		27 (25–32)	26 (22–32)
Method used to measure recency		Timing patients who sero-convert based on Fiebig stage II; IV or V	HIV-1 LAg-Avidity serological assay
Fully covered *gag* proteins (n)	P17	41	28
	P24	41	34
	P1P2	41	151
	P6P7	41	151

ETP—early time point; LTP—late time point; HIV—human immunodeficiency virus; Q1—first quartile; Q3—third quartile; VL—viral load.

## Data Availability

All relevant data are presented within the paper and the Appendix A. BCPP HIV-1 sequences and associated clinical data are available on reasonable request through the PANGEA consortium (www.pangea-hiv.org, accessed on 28 August 2020). BCPP data are available at https://data.cdc.gov/Global-Health/Botswana-Combination-Prevention-Project-BCPP-Publi/qcw5-4m9q, accessed on 28 August 2020. Tshedimoso HIV-1C 42 *gag* sequences are available on the Los Alamos HIV database under accession numbers GQ27538–GQ277569, GQ375107–GQ375128, and GQ87074–GQ871183. Tlhotlhomiso 14 HIV-1C *gag* sequences are available on special request through info@bhp.org.bw.

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
