# Peer review of "No Difference in the Prevalence of HIV-1 gag Cytotoxic T-Lymphocyte-Associated Escape Mutations in Viral Sequences from Early and Late Parts of the HIV-1 Subtype C Pandemic in Botswana"

_vaccines, 2023, doi:10.3390/vaccines11051000_

Round 1

Reviewer 1 Report

Comments to Author 

In this study the authors aimed at estimating the prevalence of CTL escape mutations in HIV-1C gag sequences in Botswana at different time points about 10 years apart and thus understand how the virus evolves.

For the study 196 recently HIV-infected patients were retrospectively included into the cross-sectional investigation. The presence of recent infection in all patients was confirmed by several adequately selected methods.

The authors showed that, despite the high frequencies of HLA*B57/B58 alleles in Botswana, which could affect the fitness of the virus, in reality, evolution almost does not occur, and the frequency of CTL escape mutations in both groups was quite low; some differences between groups are described. Additionally, the authors observed escape variants that were not classified in the Los Alamos database yet.

In general, the article contains data confirming that HIV evolves at a slower rate in a population level in Botswana. The authors set the task to identify CTL escape mutations that are associated with a high fitness cost.

Comments

1.    One of the shortcomings of the article is that the authors limited themselves to the study of only Gag protein, and did not look at others, for example, Tat and Nef. However, one must recognize the fact that Gag protein serves as a highly immunogenic region of HIV-1.region of HIV-1.

2.    The text mentions the high prevalence of HLA*B57/B58 alleles in Botswana. It would be interesting to know what is the frequency of elite controllers there.

3.    The abbreviation VRC   should be interpreted in the text as presumably "viral replication capacity”.

4.    Figure 2 legend should be made more detailed.

5.    Table 1: reference should be made in the text to the list of documented CTL-escape mutations.

6.    Misspellings in the word Botswana should be corrected in the text.

Author Response

Comments

  1. One of the shortcomings of the article is that the authors limited themselves to the study of only Gag protein, and did not look at others, for example, Tatand Nef. However, one must recognize the fact that Gag protein serves as a highly immunogenic region of HIV-1

We acknowledge that this is one of our study limitations and we have included it as part of the study limitations in line 907-910 “Despite, our study only focusing on HIV gag protein, our study provide important data on the HIV transmitted variants that were circulating in two timepoints that are 10 years apart, however, this warrants the further investigations on other HIV proteins.”

  1. The text mentions the high prevalence of HLA*B57/B58 alleles in Botswana. It would be interesting to know what the frequency of elite controllers is there.

We thank the reviewer for this comment. We acknowledge that our study did not look have HLA data for elite-controllers to determine the prevalence of HLA*B57/58 alleles. However, this have been noted in our study as part of our study recommendations in line 901-903 “The present trends further attest to the need to molecularly characterize HLA alleles among PLWH currently in Botswana, particularly the elite controllers.”  

  1. The abbreviation VRC   should be interpreted in the text as presumably "viral replication capacity”.

This has been corrected in line 850-851 “ or partially restore their viral replication capacity(VRC)’’

  1. Figure 2 legend should be made more detailed.

Thank you this has been added, refer to figure 2 in line 722.

  1. Table 1: reference should be made in the text to the list of documented CTL-escape mutations.

This has been corrected in line 485-487 “The 196 HIV gag sequences were analysed for known mutations associated with CTL escape   documented in Table 1[7, 21-37] [38-52].’’

  1. Misspellings in the word Botswana should be corrected in the text.

Thank you, this has been corrected in the manuscript.

Reviewer 2 Report

Dear Editor,

Mokaleng et al., in this study provides insights into the genetic diversity and sequence clustering of HIV-1 subtype C over a decade, which can aid in the design of future vaccine strategies. Although the study was limited to the gag gene of HIV-1, the results indicate that the virus is evolving in response to immune pressure but at a slow rate in this population. The obtained results are important. I have few comments:

-       Introduction section: please, improve this section including some sentences about the importance and the relevance of this study before the aim.

-       Materials and methods section:

2.1. Selection of the study samples and determination of HIV recent infection: please add a schematic workflow to quickly visualized the design of the study. Additionally, are the 42 sequences downloaded from Los Alamos HIV database used as references sequences? please specify it.

2.4. Hypermutations screening in BCPP participants: please add the methods to check quality of sequences and to genotype them.

-       Phylogenetic tree, this subsection is without number, maybe is the 2.6 subsection. However, phylogenetic analysis needs to be more detailed, please insert in this subsection: the number of sequences included the final dataset, the length of the analyzed sequences and the time span of included references sequences. In the phylogenetic tree (figure 2), the authors missed genetic distance.

-       Discussion section: I suggest to add some sentences about different HIV genotype analyzed using similar analysis. The number of studies in the same fields can increase the interest of the readers. 

Kind regards

Minor editing of English language required

Author Response

  1. Introduction section: please, improve this section including some sentences about the importance and the relevance of this study before the aim.

Thank you, for the comment.  This comment has been addressed in separated sections of the introduction in Line number 251 to 260 “We hypothesised that HIV-1C is adapting to the Botswana population especially to the dominant HLA class I alleles in the population which influence HIV specific CTL responses. This would be evidenced by HIV-1 variants in recently infected individuals evolving to accumulate CTL escape mutations over time. If this is the case, this would be important to ascertain as this would inform HIV vaccine design especially CTL based vaccines. So we here sought to determine the prevalence of HIV-1C CTL escape mutations in viruses from the earlier part of the epidemic and compare that with the prevalence of these mutations in the later part of the HIV-1 epidemic in Botswana. Besides informing vaccine design, data from this project will also add to knowledge on viral evolution in relation to immune responses.”

Materials and methods section:

  1. Selection of the study samples and determination of HIV recent infection: please add a schematic workflow to quickly visualized the design of the study.

We appreciate this important comment, supplementary figure s1, s2 and 3 have been added to the manuscript, refer to line; 276-277 “Total of 42 sequences from Tshedimoso study were used as ETP and 154 sequences from Tlhotlhomiso(Figure S1) and BCPP study were used as LTP.” , lines: 281-284 “ All the 42 gag sequences were retrieved from Los Alamos HIV database under accession numbers GQ27538–GQ277569, GQ375107–GQ375128, and GQ87074–GQ871183 (Figure S1).” and lines 291-293 “). Secondly 109 ART naïve participants who tested HIV-1 negative at first visit and HIV positive on the second visit after a year were considered as recently infected(Figure S2).”

  1. Additionally, are the 42 sequences downloaded from Los Alamos HIV database used as references sequences?

. The 42 sequences from Tshedimoso were downloaded from Los Alamos were used as early time point sequences. Noted in line 276-277 “Total of 42 sequences from Tshedimoso study were used as ETP and 154 sequences from Tlhotlhomiso and BCPP study were used as LTP.”

  1. Hypermutations screening in BCPP participants: please add the methods to check quality of sequences and to genotype them.

These comments have been addressed in lines;

(a). Lines 365-368 “The adjusted hypermutations were accounted for using the cumulative number of mutations across the length of the sequences from the analyzed HIV-1 gag gene. Sequences with p-value 0.05 were considered as hypermut. The adjustment for hypermutations was performed before CTL escape mutations analysis for quality control.”

(b).    Phylogenetic tree, this subsection is without number, maybe is the 2.6 subsection. However, phylogenetic analysis needs to be more detailed, please insert in this subsection: the number of sequences included the final dataset, the length of the analyzed sequences and the time span of included references sequences. In the phylogenetic tree (figure 2), the authors missed genetic distance.

Thank you for this comment, we have addressed it by subsection phylogenetic tree 2.7 and described the length and how phylogenetic tree was performed in Line 491-499. ‘’2.7 Phylogenetic tree. The edge-trimmed MSA of HIV gag sequences (length of 786bp nt long) was used to inferred tree topology based performing maximum likelihood (ML) analyses, and the bootstrap values were set at 1000. ModelTest v.3.7 [53] was used to select the simplest evolutionary model that adequately fit the sequence data. ML tree were implemented using IQTREE [54]. Seaview tool was used to assess the architectures of the produced trees and the final tree with timed outliers was finally visualized and annotated in Figtree v1.4.3. Posterior probabilities above 0.80 were noted as statistically significant. The pairwise distance among the sequences was assessed to determine their uniqueness as part of the quality control.’’

We also included these in result to also address your comment in line 693-713 “Based on Akaike Information Criterion (AIC); the best model was FLU+G4 and Bayesian Information Criterion (BIC) was FLU+R5. A total of 146 HIV-1 C gag sequences from PLWH in Botswana were included in the analysis based on IQTREE filtering. The data was made up of 2 sets ETP and LTP samples; ETP (n= 28) were collected in 2004–2008 and LTP (n = 117) in 2012–2018. When we performed ML tree using proteins with shorter sequence length such as P17 [HIV gag (nt position 130 – 1413), thus ETP (n = 41) and LTP (n = 28)] we observed genetic intermixing of sequences from 2 time periods. The mean pairwise distances were 0.096 and 0.106 (p=0.89), for ETP and LTP respectively (Figure S2).”

  1. Discussion section: I suggest to add some sentences about different HIV genotype analyzed using similar analysis. The number of studies in the same fields can increase the interest of the readers. 

This has been addressed in line: 867-871“In a study done in Japan(1983), individuals lacking the HLA-B*51 allele exhibited a 21% prevalence of the I135X mutation. Between 1997 and 2008, the frequency of the I135X mutation increased substantially, with accumulation observed in upto 70% of HLA-B*51 negative individuals [31].”